# Polysaccharide-Based Nanomedicines Targeting Lung Cancer

**DOI:** 10.3390/pharmaceutics14122788

**Published:** 2022-12-13

**Authors:** Asif Ahmad Bhat, Gaurav Gupta, Khalid Saad Alharbi, Obaid Afzal, Abdulmalik S. A. Altamimi, Waleed Hassan Almalki, Imran Kazmi, Fahad A. Al-Abbasi, Sami I. Alzarea, Dinesh Kumar Chellappan, Sachin Kumar Singh, Ronan MacLoughlin, Brian G Oliver, Kamal Dua

**Affiliations:** 1School and of Pharmacy, Suresh Gyan Vihar University, Jagatpura, Mahal Road, Jaipur 302017, India; 2Department of Pharmacology, Saveetha Dental College, Saveetha Institute of Medical and Technical Sciences, Saveetha University, Chennai 600077, India; 3Uttaranchal Institute of Pharmaceutical Sciences, Uttaranchal University, Dehradun 248007, India; 4Department of Pharmacology, College of Pharmacy, Jouf University, Sakaka 72388, Saudi Arabia; 5Department of Pharmaceutical Chemistry, College of Pharmacy, Prince Sattam Bin Abdulaziz University, Al Kharj 11942, Saudi Arabia; 6Department of Pharmacology, College of Pharmacy, Umm Al-Qura University, Makkah 21955, Saudi Arabia; 7Department of Biochemistry, Faculty of Science, King Abdulaziz University, Jeddah 21589, Saudi Arabia; 8Department of Life Sciences, School of Pharmacy, International Medical University, Bukit Jalil, Kuala Lumpur 57000, Malaysia; 9School of Pharmaceutical Sciences, Lovely Professional University, Phagwara, Punjab 144411, India; 10Faculty of Health, Australian Research Centre in Complementary and Integrative Medicine, University of Technology Sydney, Ultimo, NSW 2007, Australia; 11Research and Development, Science and Emerging Technologies, Aerogen, IDA Business Park, Dangan, H91 HE94 Galway, Ireland; 12School of Pharmacy & Biomolecular Sciences, Royal College of Surgeons in Ireland, D02 YN77 Dublin, Ireland; 13School of Pharmacy and Pharmaceutical Sciences, Trinity College, D02 PN40 Dublin, Ireland; 14School of Life Sciences, Faculty of Science, University of Technology, Sydney, NSW 2007, Australia; 15Woolcock Institute of Medical Research, University of Sydney, Sydney, NSW 2000, Australia; 16Discipline of Pharmacy, Graduate School of Health, University of Technology Sydney, Ultimo, NSW 2007, Australia

**Keywords:** lung cancer, polysaccharide, nanomedicines, immunity, chemotherapy

## Abstract

A primary illness that accounts for a significant portion of fatalities worldwide is cancer. Among the main malignancies, lung cancer is recognised as the most chronic kind of cancer around the globe. Radiation treatment, surgery, and chemotherapy are some medical procedures used in the traditional care of lung cancer. However, these methods lack selectivity and damage nearby healthy cells. Several polysaccharide-based nanomaterials have been created to transport chemotherapeutics to reduce harmful and adverse side effects and improve response during anti-tumour reactions. To address these drawbacks, a class of naturally occurring polymers called polysaccharides have special physical, chemical, and biological characteristics. They can interact with the immune system to induce a better immunological response. Furthermore, because of the flexibility of their structures, it is possible to create multifunctional nanocomposites with excellent stability and bioavailability for the delivery of medicines to tumour tissues. This study seeks to present new views on the use of polysaccharide-based chemotherapeutics and to highlight current developments in polysaccharide-based nanomedicines for lung cancer.

## 1. Introduction

Cancer is one of the leading causes of death in the world [1]. Lung cancer develops in lung tissues, particularly in the cells lining air passageways. Small-cell lung cancer (SCLC) and non-small-cell lung cancer (NSCLC) are the two primary subtypes [2]. In 113 nations, cancer is either the top cause of death or the second highest cause of death. According to WHO figures for 2020, there were 10.0 million cancer-related deaths globally and 19.3 million new disease cases [3]. Lung cancer deaths accounted for 1.80 million fatalities in 2020 or about 18% of all cancer deaths [4,5]. At present, chemotherapy is the main course of treatment for advanced lung cancer. However, almost all conventional chemotherapy drugs share the same drawbacks that limit their efficacy, including a lack of targeting ability, poor bioavailability, drug resistance, and severe side effects, such as hair loss, nausea, and immune system impairment while treating lung cancer. As a result, different auxiliary medications are frequently utilized in clinical settings to boost chemotherapy effectiveness and lessen side effects to enhance treatment outcomes and patient adherence [6,7].

A new era of cancer nanomedicine is being ushered in by nanotechnology-based chemotherapeutics, which aim to deliver flexible payloads with favourable pharmacokinetics and take advantage of molecular and cellular targeting for increased specificity, effectiveness, and safety. With nanobiotechnology’s fast development, nanomaterials are being increasingly used in therapeutic settings to treat patients. Advanced manufacturing science and engineering are combined in nanotechnology when the goal of synthesizing nanoparticles (NPs) or assembly is to create objects that are one billionth of a metre (1–100 nm) in size [8,9]. These nanomedicines target tumour tissues with precision and deliver drugs in a regulated, effective, and long-lasting manner. Nanomaterials have gained a place in treating cancer disorders because of all these benefits [10]. A class of naturally occurring polymers known as polysaccharides possess special physical, chemical, and biological features that communicate with the immunological system to promote an improved immune response. Their structural flexibility makes it possible to create multifunctional nanocomposites with great stability and bioavailability for therapeutic drug delivery to tumour tissues [11,12].

Furthermore, it has been shown that polysaccharide-based nanomedicines are efficient at enhancing anti-cancer treatment efficacy [13]. The transporters of ATP-binding cassettes might be bypassed by polysaccharide-based NPs, which are taken up by microfold cells or tumour cells that express CD44 in excess. Many anti-cancer medications, including cisplatin, paclitaxel (PTX), and doxorubicin (DOX), have significant adverse effects when administered systemically, such as hepatotoxicity, nephrotoxicity, neurotoxicity, or hypersensitivity responses. To boost the therapeutic efficiency of these anti-cancer medications and lessen their toxicity and the likelihood of side effects, this targeting polysaccharide-based nanosystems might deliver the treatments selectively to specific cells [14,15]. Additionally, polysaccharides have been observed to induce innate anti-tumour immune systems. This review aims to emphasise current developments in nanomedicines based on polysaccharides for the treatment of lung cancer and to offer fresh viewpoints on their use.

## 2. Selection of Literature

The technical results were found via Google Scholar, PubMed, Mendeley, Science Direct, Medline, and Springer Link. Numerous terminologies were used in the literature review, both singly and collectively. “Lung cancer”, “Pathophysiology of lung cancer”, “Sodium Alginate”, “Chitosan”, “Pectin”, “Chondroitin”, and “A549 cell lines” are a few of the keywords used for literature analysis. In this article, only English journals were taken into account. Even if they were not found using the first search method, relevant articles’ references were verified.

## 3. Pathophysiology of Lung Cancer

Lung cancer’s pathogenesis is extremely complicated and yet not fully understood. It is thought that a person’s chance of acquiring lung cancer is influenced by their sensitivity to specific agents and exposure to such chemicals (either occupationally or in the environment) [16,17]. Lung epithelial dysplasia is brought on by repeated exposure to pollutants such as cigarette smoke. The most frequent occupational risk factor for lung cancer is asbestos exposure. According to studies, radon exposure causes 10% of lung cancer cases, compared to outdoor air pollution, which may be responsible for 1% to 2% of instances [18]. If the encounter is prolonged, protein synthesis and genetic makeup modifications occur. As a result, the cell cycle is disrupted, which promotes cancer growth. The genetic mutations most often associated with lung cancer development include BCL2, p53 and MYC for SCLC, and KRAS, EGFR, and p16 for NSCLC [19]. They are either oncogenes or tumour suppressor genes.

Additionally, it has been demonstrated that non-smoking lung conditions, such as tuberculosis (TB), idiopathic pulmonary fibrosis, and chronic obstructive pulmonary disease (COPD), are all linked to an increased risk of lung cancer [20]. Radiation used to treat cancers other than lung cancer, especially breast cancer and non-Hodgkin lymphoma, is another risk factor [21,22]. In addition to polycyclic aromatic hydrocarbons and metals such as arsenic, nickel, and chromium, exposure to polycyclic aromatic hydrocarbons is associated with an increased risk of developing lung cancer [23] (Figure 1).

## 4. Role of Polysaccharide-Based Nanomedicines in Lung Cancer

Polysaccharides are interesting polymers composed of monosaccharides or disaccharides linked together by glycosidic bonds. Natural products, such as mannan, dextran, chondroitin, glucomannan, guar gum, chitosan, xanthan gum, alginate, and pectin, are common sources of polysaccharides. Safe, biodegradable, and highly malleable polysaccharide conjugates allow for developing novel, intelligent biological materials [24,25]. To enhance the therapeutic efficacy of lung cancer treatments, polysaccharide-based nanomedicines are effective. NPs made from polysaccharides have the potential to avoid being broken down by ATP-binding cassette transporters and to be taken up by their intended cells [26,27]. Compared to the systemic administration of several anti-cancer medications, including doxorubicin, paclitaxel, and cisplatin, the targeted delivery of these agents using a polysaccharide-based nanosystem might improve therapeutic effectiveness while decreasing toxicity and adverse effects [28].

Further research into polysaccharide-based nanomedicines for lung cancer immunotherapy is prompted by reports that specific polysaccharides may boost intrinsic anti-tumour immune systems. Several polysaccharides have been investigated as immune adjuvants and drug delivery platforms for immunotherapeutic drugs in treating lung cancer [29,30]. These include chitosan, hyaluronic acid, dextran, sodium alginate, chondroitin, and pectin (Table 1).

### 4.1. Chitosan-Based Nanomedicines

Deacetylation of chitin, a natural polymer that is abundantly present in the shells of arthropods and the cell walls of fungi, produces chitosan, a positively charged compound [13]. Due to its amino groups’ strong protonation capacity in low pH environments, chitosan is acid-soluble. It may be preferable for “pH-responsive manners in lysosomes and endosomes, two acidic subcellular organelles. Chitosan and its oligomers have been shown to interact with negatively charged substances, such as tripolyphosphate (TPP), to create a range of nanosystems with different particle sizes and zeta potentials [49]. Since it can be used to develop nanosystems with various functions and has good biocompatibility, non-toxicity, and mucoadhesiveness for increasing drug absorption and bioavailability, chitosan is frequently regarded as one of the most commonly used polysaccharides in the field of nanomedicine [50]. Chitosan-based nanomedicines often exhibit great biocompatibility, bioactivity, and in addition to their strong biodegradability, they are polycationic [51]. Therefore, they are efficient nano-carriers for medication delivery applications [52]. As further discussed below, various research has recently been conducted to provide nanomedicines for cancer treatment using chitosan-based nanomaterials. Chitosan-based epithelial growth factor receptor (EGFR)-targeted bioadhesive nanomedicine was created by Viswanadh et al. to distribute docetaxel (DTX) via the quality by design (QbD) method. The desired-size nanoparticles, which could maintain drug release for up to 70 h, were produced. The pharmacokinetic analysis demonstrated the produced NPs’ good systemic bioavailability. After a 24 h incubation period, the cytotoxicity of produced NPs was investigated in human lung adenocarcinoma A549 cells compared to free DTX. The results showed that Docetaxel-tocopherol polyethene glycol 1000 succinate-poly(lactide) nanoparticles (DTX-CS-TPGS-NPs) induced more cytotoxicity than DTX in in vitro experiments [53]. Arya et al. created composite poly(d,l-lactide-co-glycolide)-chitosan particles to treat lung cancer. This study established concentration-dependent toxicity on an NSCLC line, NCI-H460, following independent exposure to the two medicines (free-state). Paclitaxel (Tx) and Topotecan (TPT) treatment to NCI-H460 cells simultaneously showed antagonistic effects. On the other hand, synergism was demonstrated with sequential delivery, which involved preexposure to TX, followed by a drug-free incubation period and then exposure to TPT. The NCI-H460 cell viability was further decreased by TPT-loaded chitosan micro-/nanoparticles with longer incubation times, showing that TPT bioactivity was unaffected by the high solvent/voltage conditions utilised in the electrospraying procedure. The newly developed formulation incorporating Tx- and TPT-loaded composite particles demonstrated synergism when tested on an in vitro 3-D tumour model as contrasted to the synergistic effect of Tx (free-state) and TPT (PLGA-coated TPT-loaded chitosan composite particles) when treated separately [54]. Cirillo et al. produced multi-walled carbon nanotubes coated in chitosan to serve as a pH-responsive transporter to vectorise methotrexate to lung cancer. The survival of both cell lines was shown to be unaffected by empty CS multi-walled carbon nanotubes (MWCNT), demonstrating the importance of great biocompatibility for any carrier device. The free drug demonstrated the anticipated concentration-dependent cytotoxicity, reducing cell viability in non-small-cell lung carcinoma (NSCLC) H1299 cell line and healthy MRC-5 cell line instances to 59% and 50%, respectively, at the highest measured concentration. The MRC-5 cells’ unique metabolic characteristics and great susceptibility to nearly any chemical species are attributed to the drug’s identical effects on both cell lines. The treatments did not significantly impact the vitality of healthy MRC-5 cells at the measured doses, which is an interesting finding about the loaded methotrexate MTX@CS MWCNT. Additionally, it was discovered that MTX@CS MWCNT had at least as much action against cancer cells as the free medication. When free MTX was utilised as therapy, the H1299 viability was decreased by 15%, whereas the loaded medication markedly enhanced the number of dead cells up to 44%. The effectiveness of the CS MWCNT nanohybrid in vitro was amply demonstrated by these results, which were connected to the various metabolic rates of cancerous and healthy lung cells and the various pH values of the two habitats differentially affected the MTX release [55]. Paclitaxel (PTX)-loaded liposomes containing chitosan oligosaccharide (CSO) modification to treat lung cancer were effectively prepared by Miao et al. [29]. CSO-modified liposomes increased PTX release in the modelled endosome/lysosome environment while reducing PTX leakage in the simulated bodily fluid. CSO-modified liposomes were created using synthetic CSO-coupled Pluronic P123 polymers with grafting quantities of CP50 and CP20 (CP50-LSs and CP20-LSs). Compared to P-LSs and CP20-LSs, CP50-LSs demonstrated greater cellular uptake and broad distribution in A549 cells with elevated CSO alterations, which resulted in a considerable reduction of A549 cell growth in vitro. This could be explained by the fact that a significant quantity of the tumour’s abundant collagen was deposited on the CP50-LSs, increasing the reactivity of liposomes with cells and encouraging liposome internalisation by A549 cells. PTX@CP50-LSs displayed enhanced tumour formation and better therapeutic efficacy in the A549 malignant cells mice model, with a tumour inhibition rate of almost 90% following injection. These results showed that among the examined liposomes, PTX@CP50-LSs had the best therapeutic effectiveness and the least amount of toxicity [29]. Najmi and others SiRNA/DOX-loaded chitosan-based nanoparticles were created, characterised, and tested on the A549 cell line with lung cancer. This work aimed to produce and characterise in vitro how distinct pharmacological groups of carboxymethyl dextran trimethyl chitosan nanoparticles (CMDTMChiNPs) affected gene expression, cell line migration, and apoptosis in A549 cells. To encapsulate the CMDTMChiNPs, siRNA, DOX, or siRNA-DOX were used. Then, using MTT assay and real-time PCR, the impacts of HMGA2 siRNA and DOX co-delivery were evaluated in A549 survival and molecular targets (HMGA2, E-cadherin, vimentin, and MMP9), respectively. Additionally, the capacity of the created NPs to induce apoptosis and their anti-migratory properties were examined using flow cytometry and tissue repair tests. The most effective drug formulations for A549 cell cytotoxicity, changing EMT indicators, inducing apoptosis, and inhibiting migration, were NPs loaded with DOX and siRNA [56]. Ma et al. created and synthesised a brand-new Bigua-CS dual biomolecular functionalized core–shell type magnetic nanocomposite (Ag/Bigua-CS@Fe_3_O_4_) to treat human lung cancer. By applying different concentrations of the Ag/Bigua-CS@Fe_3_O_4_ nanocomposite to the afflicted (LC-2/ad), PC-14, and HLC1 cancer cell lines by MTT assay, its cytotoxicity was examined. Consequently, the PC-14 cell line showed the greatest cytotoxicity findings and anti-human lung cancer potential outcomes of the catalyst [57] (Figure 2). 

### 4.2. Hyaluronic Acid-Based Nanomedicines

Hyaluronic acid (HA) is a naturally existing polyanionic polysaccharide in several bodily fluids, including synovial fluid and extracellular matrix. It is a repetitive disaccharide molecule comprising N-acetyl-D-glucosamine and D-glucuronic acid that makes up a linear polymer. HA has a chain length ranging from 2000 to 25,000 monomer units and a molecular weight typically between 106 and 107 Da [59]. The HA synthases, which successively join the D-glucuronic acid and N-acetyl-D-glucosamine units by alternating 1,3 and 1,4 connections, produce it at the inner face of the plasma membrane. The human body has large amounts of HA in different places, including the skin, the vitreous body of the eye, synovial fluid, and the umbilical cord [60]. In recent years, the field of customised HA production has seen the use of genetic and metabolic engineering. The regulated artificial production of HA by enzymes is a different strategy. The ease of downstream processing and lower danger of viral contamination are benefits of employing microbial and enzymatic biosynthesis for HA generation. HA is produced commercially using two basic processes [61]: extraction from animal tissues and bacterial strain-based microbial fermentation. For biological and aesthetic applications, both processes provide polydisperse high-molecular-weight HA (polydispersity range from 1.2 to 2.3) [62]. The extracellular matrix’s structure and organisation are significantly influenced by HA, which controls several cellular processes, such as cell division, migration, and proliferation [63]. To assess the effectiveness of targeting lung cancer cells in vitro and in vivo, Jeannot et al. created novel hyaluronan-based nanoparticles targeting CD44 receptors of two distinct sizes. The cellular absorption of the nanoparticles was dose-dependent and particular to CD44 and other hyaluronan receptors. They showed that, at the CD44 level, the size of hyaluronan-based NPs plays a crucial impact on cellular absorption. The cellular absorption of the nanoparticles was dose-dependent and particular to CD44 and other hyaluronan receptors. They showed that, at the CD44 level, the size of hyaluronan-based NPs plays a crucial impact on cellular absorption. Smaller-sized NP was bound and internalised in NSCLC cells more effectively than larger-sized NP. Direct delivery of these NPs into the airways did not increase their absorption by the tumours, according to in vivo investigations. They showed that the size of hyaluronan-based nanoparticles significantly influences uptake efficiency and biodistribution. Small nanoparticles actively targeted tumours that overexpressed the CD44 protein, indicating that they may be employed as a medication delivery mechanism [64]. HA-modified selenium nanoparticles were created by Zou et al. to increase the therapeutic effectiveness of PTX in treating lung cancer. The current study created a tumour-targeted nanoparticle called HASe@PTX to treat NSCLC in A549 cells. In A549 cells, HA-Se@PTX demonstrated great cellular uptake and quicker PTX release from nanoparticles at an acidic pH, simulating the acidic microenvironment of cancer cells. A549 cell proliferation, migration, and invasion were all significantly reduced by HA-Se@PTX, and A549 cell death was induced. Furthermore, HA-Se@PTX demonstrated higher anti-cancer activity in vivo than free PTX and Se@PTX [65]. Li and colleagues made zirconium phosphate nanoparticles with HA modifications for possible lung cancer treatment. The MTT assay was used to assess the findings of cytotoxicity. A cytotoxicity test of the nanoparticles without PTX was also conducted as a preliminary assessment. Over 90% of the cells lived at the highest dosage when exposed to zirconium phosphate (ZP) and HA-ZP NPs. Because of their low cytotoxicity, ZP and HA-ZP NPs have various potential uses in biomedicine and cancer treatment. PTX-loaded ZP nanoparticles demonstrated some cytotoxicity toward A549 cells compared to PTX alone. Additionally, the cytotoxicity of PTX-loaded HA-ZP NPs increased and was even on par with that of PTX alone. Additionally, the results showed that mortality increased with increasing PTX concentrations, indicating a dose-dependent action of PTX-loaded HA-ZP NPs in vitro. The enhanced permeability and retention (EPR) effect and improved PTX cellular uptake facilitated by the CD44 receptor, which combined may explain the powerful cancer targetability of HA-ZP NPs, may be the causes of higher cytotoxic effects of PTX-loaded HA-ZP NPs [66]. Parashar et al. developed HA-decorated naringenin (NAR) nanomaterials for lung cancer’s chemopreventive and therapeutic potential. The improved anti-cancer impact of NAR-HA@CH-PCL-NP with a safe profile on macrophages was shown in the cell cytotoxicity experiments on A549 cells and J774 macrophage cells. A549 cell uptake research favoured promoting increased medication absorption by cancer cells. Cell cycle arrest research (using the A549 cell line) proved NAR-HA@CH-PCL-better NP’s cytotoxic impact and active targeting. Following the discovery that NAR-HA@CH-PCL-NP has a tumour growth inhibitory impact on lung cancer caused by urethane in an in vivo model, researchers conclude that the formulation has a bright future as a therapeutic and chemopreventive drug for lung carcinoma [67]. Kumar et al. synthesised HA and dihydroartemisinin (DHA) combination and described it for in vitro testing in lung cancer cells. The in vitro cytotoxicity capability of DHA and HA-DHA attached NPs was investigated in the A549 cell line utilising the Cell Counting Kit-8 (CCK-8) assay. Compared to DHA, the cytotoxicity of HA-DHA conjugate NPs at various concentrations was assessed. The outcome showed that the cytotoxic properties of HA-DHA were better than those of free DHA. The improvement in cytotoxicity may be due to the enhanced uptake of the HA-DHA complex into cells via CD44-mediated endocytosis. This process occurs with HA conjugates [68]. To provide CD44-targeted transport in urethane-induced lung cancer, Parashar et al. produced HA-functionalized PCL NPs. The HA-PCL-CAP NPs demonstrated improved CAP cytotoxic, antiproliferative, and apoptotic properties in the A549 cell line. They also demonstrated promise in treating NSCLC. Compared to ordinary CAP and PCLCAP NPs, HA-PCL-CAP NPs showed a considerable reduction in tumour formation and recovery of oxidative stress indicators [69].

### 4.3. Alginate-Based Nanomedicines

Brown algae, often known as seaweeds or Phaeophyceae, including Laminaria japonica, Macrocystis pyrifera, Ascophyllum nodosum, Laminaria Digitata, and Laminaria Hyperborea, are sources of alginate, a naturally occurring anionic polysaccharide. Other names for alginate include algin and alginic acid [70]. Regular blocks co-polymerized with D-mannuronic acid (M) and L-guluronic acid (G) residues make up the molecular structure of alginate [71]. Typically, water-insoluble alginic acid is used to extract the economically beneficial alginate from the cell walls of brown algae. Then, these salts are transformed into soluble and pure sodium alginates [72]. Alginates have special, non-toxic, valuable qualities, such as the capacity to thicken and gel, absorb and retain water, and raise the viscosity of liquids [73]. Alginate-based nano-carrier systems have been successfully produced using a variety of preparation techniques. Numerous considerations must be considered while selecting an optimal alginate-based nanoformulation. In general, choosing the appropriate alginate-based nanoformulation may be guided by the kind of molecules or therapeutic agents being loaded, the purpose of the nano-delivery system, and how the nanomaterial will be used and administered [74]. Alginate nano-hydrogels can be made using a variety of techniques, including ionic crosslinking, covalent copolymerization, thermosensitive phase change (thermal gelation), cell crosslinking, free radical polymerization, and “click” chemistry [75]. To cure carcinoma on human lung cancer cell lines, Huang et al. successfully synthesised sodium alginate coated magnetite (Fe_3_O_4_/Alg-Ag NPs) nanocomposite on which silver nanoparticles (Ag NPs) were decorated. The synthesised nanocomposite’s cytotoxicity and anti-lung cancer properties on popular lung carcinoma cell lines, including NCI-H1975, NCI-H1563, and NCI-H1299 cell lines, were examined using the MTT assay. Cytotoxicity-free for the normal cell line i.e., human umbilical vein endothelial cells (HUVECs), the synthesised nanocomposite displayed extremely poor cell viability and strong anti-lung cancer activity dose-dependently against NCI-H1975, NCI-H1563, and NCI-H1299 cell lines. The 2,2-diphenyl-1-picrylhydrazyl (DPPH) test was utilised to ascertain the antioxidant characteristics of the produced nanocomposite in the presence of the control therapy, butylated hydroxytoluene.

Given that the synthesised nanocomposite inhibited half of the DPPH molecules, the material’s substantial anti-human lung cancer potential against common lung cancer cell lines may be connected to its antioxidant activities [76]. DOX-loaded colloidal silica NP-sodium alginate was created by Mishra et al. (DOX-biohybrid). Compared to free DOX uptake, A549 cells took in more than two times as much DOX from the DOX-biohybrid. Furthermore, an in vitro viability experiment revealed that the therapy of lung cancer A549 cells with DOX-biohybrid resulted in a 50% decrease in cell viability as opposed to a 12% loss with free DOX. Finally, they detailed how they used the spray drying technique to create a novel biohybrid drug delivery system that can potentially cure lung cancer [77]. Recombinant FGF-2 was encapsulated inside stable, alginate-based nanoparticles (ABNs) by Miao et al. to raise nuclear ERK1/2 content and cause lung cancer cell death. ABNs were absorbed by immortalised human bronchial epithelial cell lines (HBE1s) and A549 cell lines in culture by non-selective endocytosis.

In contrast to A549s exposed to unfilled (i.e., blank) ABNs, the intracellular injection of FGF-2 via ABNs drastically increased lactate dehydrogenase levels, demonstrating that FGF-2-ABN treatment damaged the altered cell integrity. However, treatment with FGF-2-loaded ABN did not substantially alter the nontransformed cells. FGF-2-loaded ABNs did not affect HBE1 nuclear ERK1/2 expression, but A549s had significantly greater nuclear levels of activated ERK1/2 than ABNs. By increasing nuclear ERK1/2 activation, the unique intracellular delivery technique of FGF-2 through NPs promoted cancer cell mortality [78]. Alginate crosslinked microcapsules were created by Román et al. as a viable drug delivery system (DDS) for treating human lung cancer. The H460 lung cancer cell line and controls on systems with free medicines were used for the in vitro tests of cytotoxicity. The in vitro release of microparticles test was performed for the in vitro validation investigations, and the MTT assay was used to determine the viability of the cells. These tests demonstrated that drug delivery therapy causes H460 cell plate death more quickly [79] (Figure 3). 

### 4.4. Pectin-Based Nanomedicines

Plant cell walls include a complex polymer called pectin. It is a large, high molecular weight macromolecule that may be converted into a hydrogel and create a flexible web of polymer chains. Rhamnogalacturonan II (RG-II), rhamnogalacturonan I (RGI), homogalacturonan (HG), and xylogalacturonan make up the complicated structural makeup of pectin (XG) [80]. Despite having similar properties, pectins can have a variety of structures depending on the source and extraction technique [81]. RhamnogalacturonaI (about 20–35% of the total pectin mass) is represented by one component of a pectin biopolymer. RG-I typically refers to various pectin polysaccharide types having chains made of repeated galacturonate disaccharide sequences connected by rhamnose via [→ 4)—α—D—GalpA-(1 → 2)—α—L-Rhap-(1 →]. Rhamnogalacturonan II (RG-II) differs from RG-I in appearance because its chain is made up of the HG chain rather than GalA-Rhap disaccharides. The category of biopolymers, known as substituted galacturonic, is highly diverse. Its linear chain is made up of D-GalpA residues that are connected via α—1,4 (as in HG) intercalated with additional residues [82]. Li et al. synthesised innovative silver (Au) NPs immobilised pectin-modified magnetic nanoparticles (Fe_3_O_4_/Pectin/Au) for the catalytic reduction of nitroarenes and investigation of its anti-human lung cancer activities. The MTT assay investigated the cytotoxicity of NPs with the HUVEC cancer cell line, LC-2/ad, PC-14, and HLC-1 normal cell line. In every case, the percentage of cell viability decreases as the concentration of nanocomposite increases. The PC-14 cell line demonstrated the synthesised nanocomposite’s strongest cytotoxicity findings and anti-human lung cancer potentials [83] (Figure 4). Chang et al. created a self-healing pectin/cellulose hydrogel loaded with limonin to cure lung cancer. This study examined limonin, a herbal medicine compound that inhibits the growth of lung tumours and promotes apoptosis by targeting the highly expressed TMEM16A ion channel. The hydrogels displayed fast gelation, excellent biocompatibility and ongoing release of limonene properties. Because limonin reduced LA795 cell proliferation, migration, and induced death, it significantly reduced lung adenocarcinoma development in xenografted mice and removed acute toxicity by prolonged release from the hydrogel. This limonin/hydrogel combination exhibited an acceptable anti-cancer effect. It reduced adverse effects in vivo by combining the Limonin’s anti-tumour properties and the hydrogel made of pec-CHO/CMC-extended AH’s release [84]. A pectin, guar gum, and zinc oxide (Pec-gg-ZnO) nanocomposite for Apoptotic Cell Death Induction Lung Adenocarcinomas was created by Hira et al. This work’s focus was assessing the Pec-gg-ZnO nanocomposite as a new anti-cancer agent. The cytotoxicity experiment demonstrated the Pec-gg-ZnO nanocomposite’s potential anti-cancer effectiveness against A549, Hela, and PC-3 cancer cells, while the hemolytic assay verified the nanocomposite’s biocompatibility. A cell cycle study of A549 cells after treatment with Pec-gg-ZnO nanocomposite demonstrated S-phase arrest and apoptosis induction. Furthermore, mitochondrial depolarization, ROS production, activation of caspase-3, and PARP1 revealed that Pec-gg-ZnO nanocomposite-induced apoptosis had begun [85]. 

### 4.5. Chondroitin-Based Nanomedicines

A crucial component of cartilage and connective tissues is chondroitin sulphate (CS). CS, a copolymer of sulfated N-acetyl-D-galactosamine and D-glucuronic acid at C6 and C4 of the glycosaminoglycans (GAGs) group, is found between the extracellular matrix or on the cell surface. The anaerobic bacteria ***Bacteroides thetaiotaomicron* and *Bacteroides ovatus***, which live in the large intestine, can break down CS. Due to this characteristic trait, CS is an excellent candidate to be used as a medication carrier. [86]. Chondroitin sulphate-modified and adriamycin (ADR)-preloaded hybrid nanoparticles were created by Liang et al. and investigated for lung cancer tumour-targeted treatment. Over 95% of the cells in the cytotoxic experiments were still alive, showing that CC and CS-CC were both very biocompatible and that ADR, not the carriers, was responsible for the cytotoxicity of CS-CC/ADR. ADR performed similarly to CS-CC/ADR in terms of its capacity to inhibit DNA replication within cells, which may have contributed to its significant cytotoxicity. CC/ADR, in comparison, had less of an impact on the cytotoxicity of A549 cells. A possible explanation for CS-CC/ADR showing more cytotoxicity than CC/ADR is that it was taken up by more cells [30]. To test the inhibitory effects of brusatol on the proliferation, migration, and invasion of cancer cells, Chen et al. created customised nanoparticles utilising glycosaminoglycan placental chondroitin sulphate. In the study, brusatol-loaded NPs (BNPs) or coumarin-6-loaded nanoparticles, plCSA-BP, and scrambled control peptide-bound BNPs or CNPs were made. The NPs were applied to SKOV3 ovarian cancer cells, HEC1A endometrial cancer cells, and A549 cells. When compared to other NPs, tumour cells were observed to be much more readily absorb plCSA-CNPs. They also showed that the plCSA-BNPs enhanced cancer cell apoptosis, inhibited cancer cell proliferation, invasion, and migration, upregulated BCL2-associated X protein BAX and cleaved caspase-3 levels while downregulating matrix metalloproteinase (MMP)2, MMP9, and B-cell CLL/lymphoma 2 (BCL2) levels. By controlling the BCL2, BAX, cleaved caspase-3, MMP-2, and MMP-9 pathways, the results showed the potential of brusatol delivered by plCSA-modified NPs as a chemotherapeutic agent for the targeted therapy of tumours. They suggested that it may be an efficient and secure method for treating various tumour types [87]. A lactoferrin–chondroitin sulphate nanocomplex was created by Elwakil et al. to simultaneously deliver doxorubicin and ellagic acid nanocrystals to cancerous lung cells. Spray drying was used to transform the nanocomplex into inhalable nanocomposites. Ellagic acid is released quicker in a specific order from the resultant 192.3 nm nanocomplex than doxorubicin. The nanocomplex showed enhanced cytotoxicity and internalisation into A549 cells through Tf and CD44 receptors. In animals with lung cancer, the inhalable nanocomposites showed strong anti-cancer effectiveness and deep pulmonary deposition (89.58% fine particle fraction [FPF]) [88]. Lin et al. used poly(-caprolactone)-g-chondroitin sulphate copolymers as an intracellular doxorubicin transport vehicle to treat lung cancer cells. Using an NCI-H358 xenograft nude mice model, Micelle DOX’s in vivo tumour-targeting effectiveness was demonstrated. Micelle DOX-treated mice displayed decreased growth of the NCI-H358 [89]. Garg et al. created cellulose acetate phthalate (CSAC) core shield NPs with 5-FU as their drug as anti-cancer medication. On the A549 cell line, the CSAC NPs’ ability to reduce tumour cell proliferation was evaluated using the Sulfate Reducing Bacteria (SRB) test. The obtained data demonstrated that cell viability decreased with increasing drug concentration. After incubation, the A549 cell line was exposed to 5-FU-loaded CSAC NPs, and cell growth suppression was assessed. The high concentration of the CSAC NPs system was discovered to have a higher inhibitory impact on the cytotoxicity of cell A549. The CSAC NPs display superior cytotoxic potential characteristics against A549 cells compared to the standard medication, as demonstrated by the MTT assay (Table 2) [90].

### 4.6. Combination-Based Polysaccharide Nanomedicines

#### 4.6.1. Alginate and Chitosan-Based Nanomedicines

Ak created doxorubicin-coupled PEG acid-linked alginate/chitosan nanoparticles. The A549-luc-C8 cells, designated as a model for NSCLC, were used for the cytotoxicity experiments. Without doxorubicin, the empty nanoparticles did not exhibit cytotoxicity against cells. However, DOXPEG-NP and free doxorubicin caused dose-dependent toxicity, and the cytotoxicities worsened over time. Furthermore, unlike free drug forms, DOX-PEG-NP showed somewhat higher IC50 values than free DOX, and the bond breakdown and drug release occurred more slowly and under controlled conditions [89,91]. The pharmacokinetics and in vitro anti-cancer efficacy of empagliflozin, which contains chitosan-alginate nanoparticles in an orodispersible film, were assessed by Sinha et al. Free empagliflozin in the orodispersible film showed a 2.5-fold lower fatal effect than chitosan-alginate nanoparticles of empagliflozin, as demonstrated by an in vitro cytotoxicity study on A549 lung cancer cells [92]. To overcome the issues with anti-cancer drug delivery, Singh et al. used modified single-walled carbon nanotubes (SWCNTs) as nano-carriers for curcumin administration in lung cancer cells. Covalent modifications were made to SWCNTs, and complimentary polysaccharides (ALG and CHI) were used to encapsulate them. This improved the biodistribution of the hydrophobic CUR molecules inside cancer cells while loading them with them easier.

Additionally, the CUR-loaded functionalized SWCNTs showed higher lethal effects against human lung adenocarcinoma A549 cells compared to free curcumin, demonstrating unique drug release features indicating stability under physiological settings. They concluded that the nanoscale drug carrier might be exploited for specific targeting of cancer tissues while retaining the efficacy of chemotherapeutic medications since the released CUR caused apoptotic actions on A549 cells in a dose-dependent way [93]. Chitosan-Alginate Nanoporous Carriers were created and characterised by Alsmadi et al. Cisplatin was loaded and in vivo toxicity was tested for the treatment of lung cancer. Cisplatin’s lung toxicity was decreased by loading it onto the newly created carrier. Still, its liver toxicity after intratracheal injection rose, with nephrotoxicity being proportionate to cisplatin dosage in the case of carrier-loaded cisplatin. Furthermore, in subacute trials following intratracheal delivery, loading cisplatin on the carrier decreased the death rate and stopped weight loss in rats compared to free cisplatin. The new carrier, therefore, demonstrated strong potential for cisplatin-tailored administration for inhalational lung cancer therapy (Table 3) [94]. 

#### 4.6.2. Hyaluronic Acid and Chitosan-Based Nanomedicines

Zhang et al. created and characterised HA-modified CS NPs-HA coupled with cyanine 3 (Cy3)-labelled siRNA to examine the cytotoxic and anti-cancer effects of sCS NPS-HA in vitro as well as sCS NPs-HA. The findings demonstrated that noncytotoxic micro-sized (100–200 nm) CS NPs-HA efficiently delivered Cy3-labeled siRNA to A549 cells through receptor CD44 and suppressed cell proliferation by silencing the target gene BCL2. Results from in vivo experiments showed that compared to unaltered NPs loaded with siRNA (sCS NPs) and naked Cy3-labeled siRNA, sCS NPs-HA directly transported larger quantities of Cy3-labeled siRNA to the tumour locations, inhibiting tumour development by downregulating BCL2 [98]. Hyaluronic acid (HA) and chitosan (CS) complexation was used by Almutairi et al. to extend the half-life and activity of Raloxifene (RX), and RX-HA-CS nanoparticles (NPs) were tested for their pro-apoptotic and cytotoxic effects on hepatocellular carcinoma (HCC) (HepG2 and Huh-7) and NSCLC A549 cell lines. This work is the first to evaluate RX’s effectiveness against lung cancer cells in its HA-CS nanoformulation. It showed its cytotoxic and apoptotic capabilities against the A549 cancer cell line. The findings show that raloxifene-loaded chitosan nanoparticles decorated using hyaluronic acid (RX-HA-CS NPs) dramatically reduce lung cancer A549 cell viability via elevating NO levels, which ultimately cause apoptosis [99]. Lee et al. developed a CD44 receptor targeting HA-decorated GC nanoparticle connected to DOX via a pH-sensitive linker and co-loaded with CXB to treat NSCLC. They created the hydrophobic anti-inflammatory drug HA-GC-DOX/CXB, a GC-DOX core wrapped in HA. Drugs were successfully delivered into NSCLC by HA-decorated nanoparticles via CD44-mediated endocytosis, and the GC-attached DOX core displayed a pH-sensitive sustained release in vitro. When used in vivo, HA-GC-DOX/CXB effectively stopped tumour development. In mice (murine A549-Luc xenograft model) treated with HA-GC-DOX/CXB, histological examination revealed increased DOX accumulation and striking decreases in Ki-67 and COX-2 expression in tumour tissues. Furthermore, gene and protein analyses demonstrated that HA-GC-DOX/CXB markedly increased caspase-3 while markedly decreasing NF-kB, MMP-2, and COX-2 in tumour tissues—all without manifesting any hepatotoxicity. Results showed that supplying an anti-cancer drug to the CD44 receptor target in a mixture with an anti-inflammatory drug effectively inhibited NSCLC tumour growth. A useful prodrug that enables regulated drug release in the acidic environment characteristic of tumour microenvironments is HA-GC-DOX/CXB [100].

#### 4.6.3. Chitosan and Agarose Based Nanomedicines

Cai et al. created a Fe_3_O_4_ nanocomposite supported by chitosan-agarose modifications and assessed anti-cancer research against lung and liver cancer cells. The material’s biological activity was assessed by examining its antioxidant and cytotoxic effects on lung and liver cancer cell lines. Ag/CS-antioxidant Agar@Fe_3_O_4_’s capacity was evaluated using DPPH radical scavenging experiments. Studies using Ag/CS-Agar@Fe_3_O_4_ against the HepG2 and A549 cell lines were conducted for liver and lung cancer. Without harming the normal cell line, the nanomaterial’s percentage of cell viability decreased in both cell types dose-dependently. The findings show that the Ag/CS-Agar@Fe_3_O_4_ nanocomposite is a potent chemotherapeutic agent against hepatocellular carcinoma and lung cancer cells [97]. 

## 5. Conclusions and Future Perspective

In terms of morbidity and mortality, lung cancer is one of the most malignant tumours that has increased at an alarming rate in recent years. Significant advances have been achieved in the detection, diagnosis, and treatment of lung cancer attributable to the nano-drug loading technology in recent years. In order to reduce the potential for therapy-related side effects and maximise bioavailability, nanomaterials are being employed to direct the delivery of drugs directly to tumour tissue. Passive targeting uses the enhanced penetration and retention (EPR) effect, while active targeting uses nanomaterials loaded with recognition ligands for tumour marker molecules to produce the desired effect. EPR effect has been shown to be beneficial in rats but not humans. This is why the active targeted nano-drug loading method has received so much attention from the scientific community. It has also been shown to have enhanced selectivity for cancer cells and a stronger affinity for those cells. Increasing the specific binding of NPs carrying medications to disease cells improves the effectiveness of chemotherapy, and several receptors, including VEGFR, integrin, EGFR, FR, TFR, CD44, and receptor, are capable of doing so. Apoptosis and cell death induced by ligands, for instance, are specific to malignant tumours and have no effect on normally dividing cells such as stem cells.

The carrier system for lung cancer treatment has been revolutionised by nanotechnology, opening doors to tremendous future prospects. Due to their unique combination of physicochemical and biological properties, polysaccharides have found widespread application as carrier materials in the development of various nanoparticles with diagnostic and drug delivery applications. Coating with different polymers can protect them from degradation and improve epithelial absorption, and studies have shown that nano-sized formulations can be obtained by simple ionic complexation of polysaccharides, such as chitosan and alginate, with proteins and peptides, resulting in an increased in vivo residence time. Lipophilic medicines, mostly anticancer medications, have been successfully synthesised in polysaccharide-based nanoparticles to target tumour through the EPR effect and receptor-mediated endocytosis. Clinical advancement of polysaccharide-based nanoparticles as an anticancer treatment is reliant on a deeper mechanistic knowledge of the interactions between nanoparticles and malignancies. Hybrid nanoparticles are more stable, they can be more precisely targeted, and have a longer half-life.

In terms of stability, polysaccharide-based NPs are greatly diluted after systemic administration, and their dynamic condition might pose a threat to their structural integrity. Destabilization of the NP might also result from an increase in drug loading content. When it comes to enhancing their usefulness in vivo, polysaccharide-based NPs benefit greatly from being cross-linked either ionically or covalently, since this greatly increases their structural stability. As an added bonus, mineralization of NPs can be used to transport therapeutic agents with increased stability in physiological conditions compared to bare NPs, but enhanced delivery in acidic intracellular components, such as endosomes and lysosomes, or in the slightly acidic conditions of the extracellular matrix in tumour regions.

Some polysaccharide chemistry is highly variable, making it difficult to accurately specify the delivery system, and polysaccharides may have wide and/or mixed molecular weights, both of which might be downsides when used in drug administration. A further barrier to chemical modification is the insoluble nature of many polysaccharides in common organic solvents. Finally, if quick release is required, polysaccharide-based drug delivery systems are not always feasible since delayed enzymatic breakdown of the biopolymer is necessary for prolonged release. As such, carriers that can respond to external stimuli may prove to be an advantageous upgrade over traditional polysaccharide delivery methods.

## Figures and Tables

**Figure 1 pharmaceutics-14-02788-f001:**
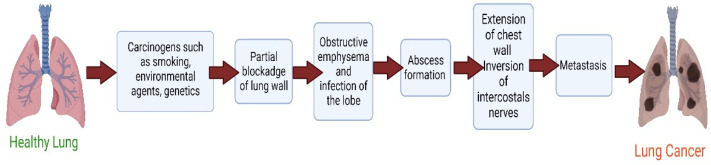
Pathophysiology of lung cancer.

**Figure 2 pharmaceutics-14-02788-f002:**
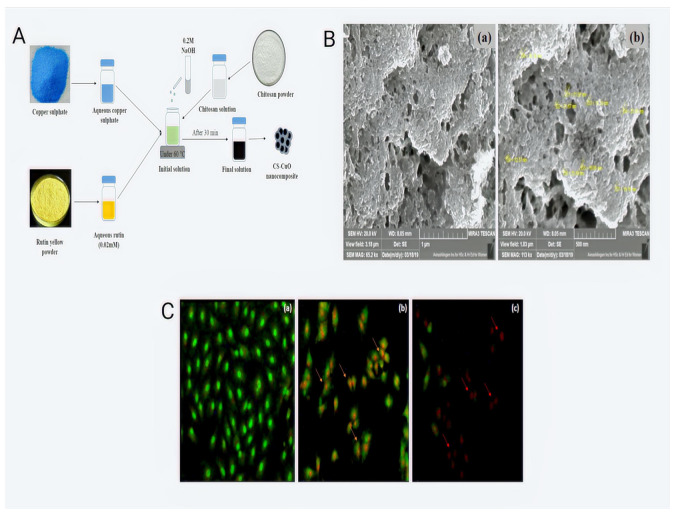
Anti-proliferative activity of chitosan/copper oxide nanocomposites containing rutin in human lung cancer cells; a systematic depiction of bio-inspired synthesis: (**A**) synthesis of the CS-CuO-rutin nanocomposite, (**B**) FE-SEM study, (**a**) CS-CuO nanocomposite at 65.2 kx magnifications (**b**) CS-CuO nanocomposite at 113 kx magnifications (**C**) apoptosis study in A549 cells using the AO/EtBr dual staining method. A549 cells that are untreated and display a green hue are alive, while orange and red hues suggest early cell death for rutin and late cell death for the CS-CuO nanocomposite (**a**) control cells, (**b**) rutin treated and (**c**) CS-CuO nanocomposite treated [58].

**Figure 3 pharmaceutics-14-02788-f003:**
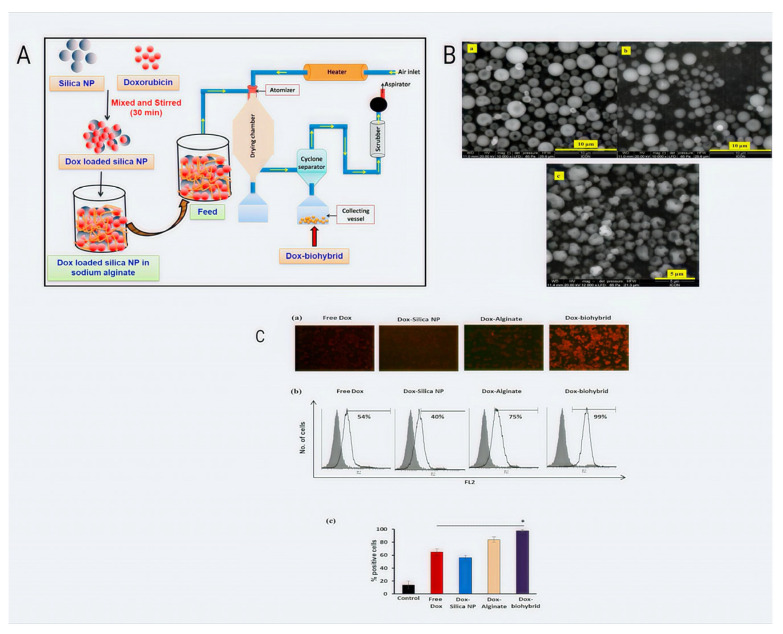
Systematic demonstration of spray drying as a successful method for producing doxorubicin drug carriers made of silica nanoparticles and sodium alginate. (**A**) Flowchart of Dox-biohybrid drug carrier manufacture. (**B**) SEM micrographs of (**a**) 1% Dox-silica nanoparticles (Dox-silica NP), (**b**) 2% Dox-sodium alginate (Dox-sodium alginate), and (**c**) a biohybrid of the two (Dox-biohybrid). (**C**) Examination of the cellular uptake of free Dox and encapsulated Dox using fluorescence microscopy. (**a**) Fluorescence microscopy image of treated cells. (**b**) Quantitative assessment of treated cells by flow cytometry. (**c**) Percent positive cells with Dox expression analyzed through flow cytometry from three independent experiments. (* *p* < 0.05) [77].

**Figure 4 pharmaceutics-14-02788-f004:**
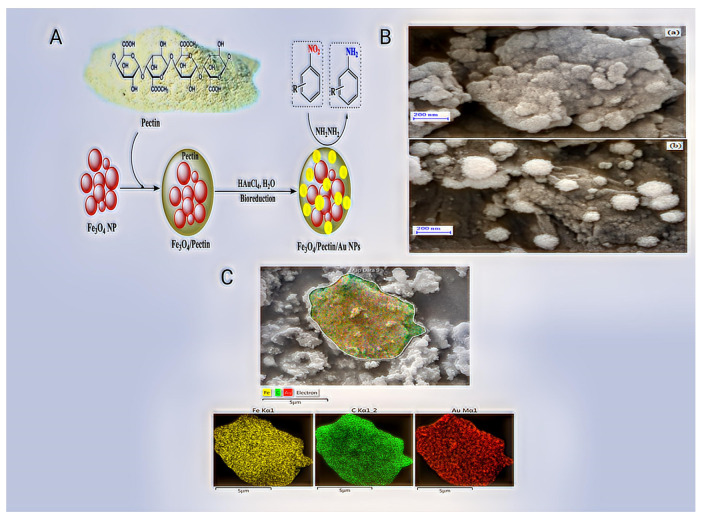
A novel magnetic nanocomposite (Fe_3_O_4_/Pectin/Au) for catalytic reduction of nitroarenes and investigation into its anti-human lung cancer properties, with a systematic representation of the in situ decorated Au nanoparticles on pectin-modified Fe_3_O_4_ nanoparticles. (**A**) Fe_3_O_4_/Pectin/Au nanocomposite preparation and use in the reduction of nitroarenes (**B**) Homogenous dispersion of Fe, C, and Au atoms over the nanocomposite surface is shown by FESEM analysis. (**a**) Fe_3_O_4_/Pectin and (**b**) Fe_3_O_4_/Pectin/Au nanocomposite. (**C**) Fe_3_O_4_/Pectin/Au nanocomposite elemental mapping [83].

**Table 1 pharmaceutics-14-02788-t001:** Nano-based lung cancer treatment using polysaccharides: structure and benefits.

Polysaccharide Type	Structure	Advantages
Chitosan	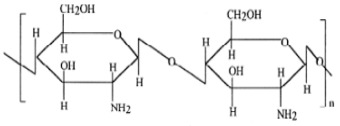	Facilitating mucoadhesion [31], Biodegradability [32], Low toxicity [32], Biocompatibility [33], Easy to prepare [34], pH-responsiveness [35].
Hyaluronic acid	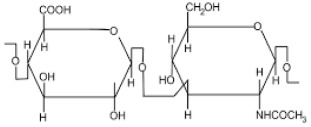	Biocompatibility [36], Biodegradability [37], No immunogenicity [38], Non-toxic [39], Strong affinity for cancer cell receptors such as CD44 [40].
Sodium Alginate	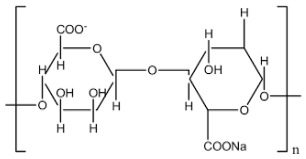	Ease of preparation, Biocompatibility, Biodegradability, Non-toxicity, Physicochemical versatility for the insertion of targeted moieties.
Chondroitin	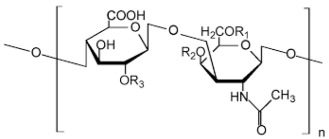	Cell adhesion [41], Biodegradation [42], Bioavailability [43], Viscoelasticity [44].
Pectin	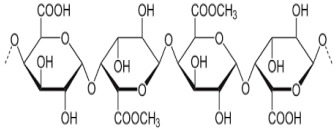	Easy availability [45], Biodegradability [46], Economic [47], Safe [48].

**Table 2 pharmaceutics-14-02788-t002:** Nanoparticles derived from polysaccharides as delivery vehicles for lung cancer treatment.

Polysaccharide Type	Nanomaterial	Loaded Agents	Therapeutic Effects	References
Chitosan	Poly(d,l-lactide-*co*-glycolide)–chitosan composite particles Multi-walled carbon nanotubes coated with chitosan	Paclitaxel and Topotecan Methotrexate	Synergism, Enhanced cell death. Selective in killing cancer cells.	[54,55]
Hyaluronic acid	Hyaluronic acid-modified selenium nanoparticles Hyaluronic acid-modified zirconium phosphate nanoparticles	Paclitaxel Paclitaxel	Impede migration, Reproduction, cell invasion of A549. The CD44 receptor mediates cellular uptake.	[66]
Alginate	Sodium alginate colloidal silica nanoparticles	Doxorubicin	A549 cells took up DOX more than twice as much from the DOX-biohybrid as from free DOX.	[49]
Chondroitin	Pectin/cellulose hydrogel	Limonin	Inhibited proliferation promoted apoptosis.	[49]
Pectin	Biohybrid drug carrier of colloidal silica nanoparticles	Doxorubicin	A549 cells took up DOX more than two times as much from DOX-biohybrid as from free DOX.	[49]

**Table 3 pharmaceutics-14-02788-t003:** Combination of polysaccharide-based nanoparticles as lung cancer treatment delivery systems.

Polysaccharide Type	Nanomaterial	Loaded Agents	Therapeutic Effects	References
Alginate and chitosan	PEG diacid-linked alginate/chitosan nanoparticles	Doxorubicin	Dose-dependent toxicity.	[95]
Hyaluronic acid and chitosan	Hyaluronic acid (HA)- modified chitosan nanoparticles	Cyanine 3 (Cy3)-labelled siRNA	Slowed down cell division markedly reduced the transcription factors BCL2.	[96]
Chitosan and agarose	Silver/chitosan-Agar nanocomposite	Fe_3_O_4_	Antioxidant potential dose-dependent toxicity.	[97]

## Data Availability

Not applicable.

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
