# Peer review of "Polysaccharide-Based Nanomedicines Targeting Lung Cancer"

_pharmaceutics, 2022, doi:10.3390/pharmaceutics14122788_

Round 1

Reviewer 1 Report

The paper provides a comprehensive overview of the recent use of polysaccharide-based materials developed and applied for lung cancer treatment. In general, the information provided is useful and provides an update on recent work in the field. I have numerous comments on both the content and the style.

To begin with, the English language and style: I found inconsistencies throughout the manuscript; some sections need to be carefully revised and some parts rewritten, but the entire manuscript would benefit from another thorough proofreading (spelling, punctuation, missing words, abbreviations, etc.). Style should be improved and be cautious about copying and pasting from references; I discovered several uses of "our" that came directly from the reference article. I tried to include as many comments as possible in the following list, but I'm sure I left some out.

For the overall organization of the review, I would reorganize the paragraph distribution, do not change paragraphs in the middle of the discussion (see lines 189, 412, 537, 603), and divide more sections 4.1, 4.2, and 4.4 into smaller paragraphs to facilitate reading.

In terms of figures, the design of the first figure could be improved. I would reduce the number of figures cited from each reference and keep only the most important that are being discussed in the text (Figures 2,3,4,5).

After discussing some examples, I would add some conclusion sentences; lines 617—619 are a good example. Similarly, I would add some concluding sentences and bridge sentences at the end of the sub-sections to make the reading flow more smoothly.

Please use a more general publication for the introduction section, such as a review on lung cancer, nanomaterials, or similar, rather than research articles (Ref 1 to 12).

For the conclusion section, consider including some questions, points to discuss, or what is lacking in the literature on:

-          Cellular uptake mechanisms of polysaccharide-based nanoparticles.

-          Clearance (studies in vivo animals and humans).

-          In vivo stability of polysaccharide-based NPs.

-          The polysaccharide-based NPs synthesis process (difficulties, cost).

-          Possibility of functionalization using specific ligands (peptides, proteins, antibodies, etc.).

Specific remarks:

-          51: encourage => induce

-          59: The top cause of death in the world is cancer => incorrect: cardiovascular diseases are. Include a citation.

-          60: SCLC and NSCLC: abbreviations are required

-          61: Change reference, use a lung cancer review for example.

-          61-62: « to see how they look » too familiar. Classification is made using histological features

-          63: Repetition: “According to WHO figures for 2020”

-          73-77: I would move the section about polysaccharides after the paragraph about the use of nanotechnology for cancer targeting and delivery.

-          86: Before introducing polysaccharide-based nanomaterials, you can include some examples of nanomaterials used in cancer treatments that are organic (liposomes, micelles, dendrimers) and inorganic (metal NP and nanoclusters, quantum dots, etc.).

-          89: “nanoparticles” introduce an abbreviation for later use (NPs)

-          121: spelling: non-smoking  + TB and COPD abbreviations

-          129: Figure legend/description is missing

-          Table 1:

o   Improve formatting (one advantage per line, consistency with punctuation, capital letters)

o   chitosan: “ less toxic” means low toxicity or less toxic than something else?

o   pectin: “Safe and devoid of side effects” is redundant.

-          155: arthropods not anthropods – Check carefully your references, you copied the mistake in ref 27, their reference (10.1016/j.carbpol.2010.04.074) used the correct term.

-          160: “TPP” abbreviation

-          167: “they” capital letter

-          172: QdB method : define/abbreviation

-          172: Use only surname: Viswanath et al.  – Same for the rest of the manuscript.

-          176: “at 37°Cnot necessary, it’s classical cell culture conditions.

-          177: “in contrast to that of DTX” I'm not sure what you meant.

-          178: DTX-CS-TPGS-NPs: abbreviation

-          179: “Studies” is it only ref 32? If not add corresponding references

-          181: “CTX” abbreviation

-          182 “Rutin” => define – CS-CuO abbreviation

-          184: spelling: anti-proliferative

-          186: You already define it line 177 that A549 are human lung adenocarcinoma cells. Same line 187, 351, 426, 536, 549, 561, …)

-          187: viability not vitality

-          191: encourages => improves – Examine the sentence's ending meaning: “making A549 cancer cells more permeable to cells and mitochondria”.

-          193-196: “The absorption of NPs by cells rises when cell/mitochondrial permeability increases, which activates caspases to cause apoptosis in the cells. The information gained revealed that the CS-CuO nanocomposite may cause apoptosis, leading to cell death”

o   Improve style, flow: “When cell/mitochondrial permeability increases, CS-CuO nanocomposite absorption increases, and the anti-proliferative action promotes caspase activation, triggering apoptosis and leading to cell death."

-          197: cure => treat

-          199: Tx and TPT abbreviations

-          211: add abbreviation MWCNT

-          217: Describe the different types of cell lines: H1299 and MRC-5.

-          248: “MSeifi-Najmi and others created, characterized, and tested SiRNA/DOX-loaded chitosan-based nanoparticles on the A549 cell line” add the abbreviation for DOX line 91.

-          252: consistency with the same abbreviation A549 not A-549

-          253: MTT assay

-          267: copy paste from reference “our”

-          Figure 2: figures from ref 33 are too small and pictures are distorted

-          288: word missing “amounts of HA in X places”

-          312: uptake/internalization instead of “entrapment”

-          323: Add ZP abbreviation

-          334: “The enhanced penetration and retention” add ref for the EPR effect and it is permeability instead of penetration

-          344: spelling: line

-          348-349: Review sentence “Robin Kumar and others X Hyaluronic acid and….”

-          406: “NCI-H1298“ should be NCI-H1299.

-          410: acronyms (HUVECs) should be line 407

-          421: spelling: detailed

-          433: “our”

-          436:”DDS” abbreviation

-          438: “Petri plates were used” is not worth mentioning.

o   The in vitro release of microparticles test was performed for the in vitro validation investigations, and the MTT assay was used to measure the viability of the cells.

-          439: viability not vitality

-          467-468: In the same sentence, the words created and synthesized are redundant.

-          469: This sentence can be combined with the one before it.

-          467-471: All three sentences should be changed. It's as if the reference's title was separated into three sentences.

-          471-474: incorrect. The MTT test does not evaluate interactions.

-          476: “our”

-          516: Remove “It is often found.”

-          517: “Bacteroidesthetaiotaomicron and Bacteroidesovatus » spelling

-          519: « we »

-          519-522: To double-check, the reference does not correspond to the text.

-          534: “This investigation created… “ to modify

-          534-544: Section needs to be rewritten.

-          560: “SBR” abbreviation

-          565-567:  “to findings in statistical evidence” to remove.

o Simplify “…. when compared to the standard medication, as demonstrated by the MTT assay (Figure 6) (Table 2)[67].”

-          Tables 2 and 3:

o   Improve formatting (consistency with punctuation, capital letters)

-          628-632: Did the IC50 was reduced in all cell lines? Explain the findings, are there any differences between cancer types? Is the lung cancer model different?  Remove “according to the data”.

-          646: HCC abbreviation

-          649: When you write specific, is it also toxic for HCC cell lines? if so, is it more toxic for lung cancer cells?

-          659: what kind of mice? (murine A549-Luc xenograft model)

-          669: Spelling : Chitosan

-          690 and 696: “The” spelling capital

Author Response

Author reply to reviewer 1 comment

The paper provides a comprehensive overview of the recent use of polysaccharide-based materials developed and applied for lung cancer treatment. In general, the information provided is useful and provides an update on recent work in the field. I have numerous comments on both the content and the style.

To begin with, the English language and style: I found inconsistencies throughout the manuscript; some sections need to be carefully revised and some parts rewritten, but the entire manuscript would benefit from another thorough proofreading (spelling, punctuation, missing words, abbreviations, etc.). Style should be improved and be cautious about copying and pasting from references; I discovered several uses of "our" that came directly from the reference article. I tried to include as many comments as possible in the following list, but I'm sure I left some out.

For the overall organization of the review, I would reorganize the paragraph distribution, do not change paragraphs in the middle of the discussion (see lines 189, 412, 537, 603), and divide more sections 4.1, 4.2, and 4.4 into smaller paragraphs to facilitate reading.

Reply: Dear reviewer, many thanks for your valuable comments. I have corrected a paragraphs as per your comment.

In terms of figures, the design of the first figure could be improved. I would reduce the number of figures cited from each reference and keep only the most important that are being discussed in the text (Figures 2,3,4,5).

Reply: Dear reviewer, I have improved a figure 1 as per your comment. I have modified a figure (2-5). I removed a figure 6 as it was not so important.

After discussing some examples, I would add some conclusion sentences; lines 617—619 are a good example. Similarly, I would add some concluding sentences and bridge sentences at the end of the sub-sections to make the reading flow more smoothly.

Reply: Dear reviewer, I have modified a sections as per your comments.

Please use a more general publication for the introduction section, such as a review on lung cancer, nanomaterials, or similar, rather than research articles (Ref 1 to 12).

Reply: Dear reviewer, I have replaced a references 1-12 as per your comments.

For the conclusion section, consider including some questions, points to discuss, or what is lacking in the literature on:

-          Cellular uptake mechanisms of polysaccharide-based nanoparticles.

-          Clearance (studies in vivo animals and humans). 

-          In vivo stability of polysaccharide-based NPs.

-          The polysaccharide-based NPs synthesis process (difficulties, cost).

-          Possibility of functionalization using specific ligands (peptides, proteins, antibodies, etc.).

Specific remarks:

-          51: encourage => induce

Reply: I have done a correction.

-          59: The top cause of death in the world is cancer => incorrect: cardiovascular diseases are. Include a citation.

Reply: I have done a correction.

-          60: SCLC and NSCLC: abbreviations are required

Reply: I have done a correction.

-          61: Change reference, use a lung cancer review for example.

Reply: I have done a correction.

-          61-62: « to see how they look » too familiar. Classification is made using histological features

Reply: I have done a correction.

-          63: Repetition: “According to WHO figures for 2020”

Reply: I have done a correction.

-          73-77: I would move the section about polysaccharides after the paragraph about the use of nanotechnology for cancer targeting and delivery.

Reply: I have rearranged a paragraphs as per your comments.  

-          86: Before introducing polysaccharide-based nanomaterials, you can include some examples of nanomaterials used in cancer treatments that are organic (liposomes, micelles, dendrimers) and inorganic (metal NP and nanoclusters, quantum dots, etc.).

Reply: Dear reviewer, I have updated a introduction section.

-          89: “nanoparticles” introduce an abbreviation for later use (NPs)

Reply: I have done a correction

-          121: spelling: non-smoking  + TB and COPD abbreviations

Reply: I have done a correction

-          129: Figure legend/description is missing

Reply: I have done a correction

-          Table 1:

o   Improve formatting (one advantage per line, consistency with punctuation, capital letters)

o   chitosan: “ less toxic” means low toxicity or less toxic than something else?

  • pectin: “Safe and devoid of side effects” is redundant.

Reply: I have done a correction

-          155: arthropods not anthropods – Check carefully your references, you copied the mistake in ref 27, their reference (10.1016/j.carbpol.2010.04.074) used the correct term.

Reply: I have done a correction

-          160: “TPP” abbreviation

Reply: I have done a correction

-          167: “they” capital letter

Reply: I have done a correction

-          172: QdB method : define/abbreviation

Reply: I have done a correction

-          172: Use only surname: Viswanath et al.  – Same for the rest of the manuscript.

Reply: I have done a correction

-          176: “at 37°C“ not necessary, it’s classical cell culture conditions.

Reply: I have done a correction

-          177: “in contrast to that of DTX” I'm not sure what you meant.

Reply: I have done a correction

-          178: DTX-CS-TPGS-NPs: abbreviation

Reply: I have done a correction

-          179: “Studies” is it only ref 32? If not add corresponding references

Reply: Yes its only a 32 ref.

-          181: “CTX” abbreviation cancelled

Reply: I have done a correction

-          182 “Rutin” => define – CS-CuO abbreviation

Reply: I have done a correction

-          184: spelling: anti-proliferative

Reply: I have done a correction

-          186: You already define it line 177 that A549 are human lung adenocarcinoma cells. Same line 187, 351, 426, 536, 549, 561, …)

Reply: I have done a correction

-          187: viability not vitality

Reply: I have done a correction

-          191: encourages => improves – Examine the sentence's ending meaning: “making A549 cancer cells more permeable to cells and mitochondria”.

Reply: I have removed a sentence from the manuscript.

-          193-196: “The absorption of NPs by cells rises when cell/mitochondrial permeability increases, which activates caspases to cause apoptosis in the cells. The information gained revealed that the CS-CuO nanocomposite may cause apoptosis, leading to cell death”

Reply: I have removed a sentence from the manuscript.

  • Improve style, flow: “When cell/mitochondrial permeability increases, CS-CuO nanocomposite absorption increases, and the anti-proliferative action promotes caspase activation, triggering apoptosis and leading to cell death."
  • Reply: I have removed a sentence from the manuscript.

-          197: cure => treat

Reply: I have done a correction.

-          199: Tx and TPT abbreviations

Reply: I have done a correction.

-          211: add abbreviation MWCNT

Reply: I have done a correction.

-          217: Describe the different types of cell lines: H1299 and MRC-5.

Reply: I have done a correction.

-          248: “MSeifi-Najmi and others created, characterized, and tested SiRNA/DOX-loaded chitosan-based nanoparticles on the A549 cell line” add the abbreviation for DOX line 91.

Reply: I have done a correction.

-          252: consistency with the same abbreviation A549 not A-549

Reply: I have done a correction.

-          253: MTT assay

Reply: I have done a correction.

-          267: copy paste from reference “our”

Reply: I have done a correction.

-          Figure 2: figures from ref 33 are too small and pictures are distorted

Reply: I have done a correction.

-          288: word missing “amounts of HA in X places”

Reply: I have done a correction

-          312: uptake/internalization instead of “entrapment”

Reply: I have done a correction

-          323: Add ZP abbreviation

Reply: I have done a correction

-          334: “The enhanced penetration and retention” add ref for the EPR effect and it is permeability instead of penetration

Reply: I have added the reference and done the correction

-          344: spelling: line

Reply: I have done a correction

-          348-349: Review sentence “Robin Kumar and others X Hyaluronic acid and….”

Reply: I have reviewed the sentence and done a correction

-          406: “NCI-H1298“ should be NCI-H1299.

Reply: I have done a correction

-          410: acronyms (HUVECs) should be line 407

Reply: I have done a correction

-          421: spelling: detailed

Reply: I have done a correction

-          433: “our”

Reply: I have done a correction

-          436:”DDS” abbreviation

Reply: I have done a correction

-          438: “Petri plates were used” is not worth mentioning.

Reply: I have done a correction

o   The in vitro release of microparticles test was performed for the in vitro validation investigations, and the MTT assay was used to measure the viability of the cells.

-          439: viability not vitality

Reply: I have done a correction

-          467-468: In the same sentence, the words created and synthesized are redundant.

Reply: I have done a correction

-          469: This sentence can be combined with the one before it.

Reply: I have done a correction

-          467-471: All three sentences should be changed. It's as if the reference's title was separated into three sentences.

Reply: I have done a correction

-          471-474: incorrect. The MTT test does not evaluate interactions.

Reply: I have done a correction

-          476: “our”

Reply: I have done a correction

-          516: Remove “It is often found.”

Reply: I have done a correction

-          517: “Bacteroidesthetaiotaomicron and Bacteroidesovatus » spelling

Reply: I have done a correction

-          519: « we »

Reply: I have done a correction

-          519-522: To double-check, the reference does not correspond to the text.

Reply: I have done a correction

-          534: “This investigation created… “ to modify

Reply: I have done a correction

-          534-544: Section needs to be rewritten.

Reply: I have done a correction

-          560: “SBR” abbreviation

Reply: I have done a correction

-          565-567:  “to findings in statistical evidence” to remove.

o Simplify “…. when compared to the standard medication, as demonstrated by the MTT assay (Figure 6) (Table 2)[67].”

Reply: I have done a correction

-          Tables 2 and 3:

o   Improve formatting (consistency with punctuation, capital letters)

Reply: I have done a correction

-          628-632: Did the IC50 was reduced in all cell lines? Explain the findings, are there any differences between cancer types? Is the lung cancer model different?  Remove “according to the data”.

Reply: Dear reviewer, this study was creating a confusion so I have removed this study in revised manuscript.

-          646: HCC abbreviation

Reply: I have included a full form.

-          649: When you write specific, is it also toxic for HCC cell lines? if so, is it more toxic for lung cancer cells?

Reply: I have done a correction.

-          659: what kind of mice? (murine A549-Luc xenograft model)

Reply: I have done a correction.

-          669: Spelling : Chitosan

Reply: I have done a correction.

-          690 and 696: “The” spelling capital

Reply: I have done a correction.

The term ‘narcotic’ in the legal sense is quite different from that used in the medical context which denotes a sleep inducing agent. Legally, a narcotic drug could be an opiate (a true narcotic), cannabis (a non-narcotic) or cocaine (the very antithesis of a narcotic, since it is a stimulant). The term ‘psychotropic substance’ denotes mind-altering drugs such as Lysergic Acid Diethylamide (LSD), Phencyclidine, Amphetamines, Barbiturates, Methaqualone, and designer drugs (MDMA, DMT, etc.).

Reviewer 2 Report

Some specific comments are as follows:

Page 2, Line 60…..…. small cell lung cancer (SCLC) and non-small cell lung cancer (NSCLC)……… add the full name for the first time before their abbreviations.

Page 2, Line 63…..…. according to the WHO data for 2020……..Where is its’ reference?

Page 3, Line 88…..…. the transporters……The (begin with Capital letter).

Page 3, Lines 88-90……need ref., is it ref [10]?

Page 3, Line 93…..…. References number [9] and [10] have no relation to their cited paragraph about cisplatin, paclitaxel, and doxorubicin and their toxicity….Please check and add the right refs.

[9] Huang, Y.; Hu, X.; Zhao, H.; He, D.; Li, Y.; Yang, M.; Yu, Z.; Li, K.; Zhang, J. Composite alkali polysaccharide supramolecular nanovesicles improve biocharacteristics and anti-lung cancer activity of natural phenolic drugs via oral administration. Int J Pharm 2020, 573, 118864, doi:10.1016/j.ijpharm.2019.118864.

[10] Jeannot, V.; Mazzaferro, S.; Lavaud, J.; Vanwonterghem, L.; Henry, M.; Arboléas, M.; Vollaire, J.; Josserand, V.; Coll, J.L.; Lecommandoux, S.; et al. Targeting CD44 receptor-positive lung tumors using polysaccharide-based nanocarriers: Influence of nanoparticle size and administration route. Nanomedicine 2016, 12, 921-932, 761 doi:10.1016/j.nano.2015.11.018.

I thought there is something wrong in citation! or there is a mistake in references numbering? Please check the reference list with their citation within the text.

Page 3, Lines 109-113…..…. References number [13] and [14] have no relation to their cited paragraph ….Please check and add the right refs.

Page 3, Lines 116-120…..…. References number [16] and [17] have no relation to their cited paragraph ….Please check and add the right refs.

Page 3, Lines 121,122……need reference

 Page 3, Line 122…..…. COPD ……… add the full name for the first time before its abbreviation.

Page 4, Lines 124-127…..…. Reference number [20] have no relation to its cited paragraph ….Please check and add the right refs.

Page 6…..Table 1, where are the references of the advantages of those referred Polysaccharides?

Page 7, Line 160…..…. TPP……… add the full name for the first time before its abbreviation.

Page 7, Line 167…..…. they are…….Capital (They)

Page 7, Line 171…..…. EGFR……… add the full name for the first time before its abbreviation.

Page 7, Line 172……………Matte Kasi Viswanath et al.? This reference was not found in the reference partition. Is it Viswanadh [32]?

Page 7, Line 172……………QbD method……… add the full name for the first time before its abbreviation.

Page 7, Line 183…………… DevarajBharathi et al………………… Bharathi et al. [33]

Page 8, Line 196…………… AryaN et al…………. Arya et al. [34]

Page 8, Line 199…………… Tx and TPT……… add the full name for the first time before its abbreviation.

Page 8, Line 211…………… Giuseppe Cirillo and colleagues…………. Giuseppe Cirillo and colleagues [35]

Page 9, Line 232………… Yun-qiu Miao et al……….Reference number[]

Is there any explained mechanism for Chitosan based nanomedicines toxicity toward cancer cells?

Figure 2: in page 10……Those figures are from article [33], Did the author obtain permission from the copyright holder to publish under the CC-BY license? According to Publication Ethics Statement of MDPI

https://www.mdpi.com/journal/information/instructions

Page 10, Line 281……………………. HA……. add the full name for the first time before its abbreviation.

Page 11, Line 301………… VictorJeannot et al…………….. Jeannot et al [Reference number]

Page 12, Line 338………… PoonamParashar et al…………… Parashar et al. [Reference number], Lines 338-339 should be in Lowercase sentence.

Figure 3: in page 10……Those figures are from article [43], Did the author obtain permission from the copyright holder to publish under the CC-BY license? According to Publication Ethics Statement of MDPI

https://www.mdpi.com/journal/information/instructions

Page 13, Lines 383-385……….. Laminaria japonica, Macrocystis pyrifera, Ascophyllum nodosum, Laminaria digitata, and Laminaria Hyperborea………………Should be Italics

Page 16, Lines 459-460 ……. RhamnogalacturonaI About 20–35% of the total pectin mass is represented by one component of a pectin…………………………. RhamnogalacturonaI (about 20–35% of the total pectin mass) is represented by one component of a pectin

Page 16, Line 467…………… YunLi et al………….. Li et al. [Reference number]

Page 16, Line 474…………… (%).In every case,…………………….need space

Page 16, Lines 487-488………………..Pectin, Guar Gum, and Zinc Oxide Nanocomposite for Apoptotic Cell Death Induction 487 Lung Adenocarcinomas was created by Indu Hira et al……..This reference is not found in the reference part

Page 18, Lines 517……… Bacteroidesthetaiotaomicron…………correct Bacteroides thetaiotaomicron

Page 18, Lines 519-521…………………….. Recently, we produced chondroitin sulfate-based nano………..We??? I didn’t find reference for author’s work… please clarify or correct.

Page 18, Line 526……………… Adr,…………………….. add the full name for the first time before its abbreviation.

-Please check references well inside the text and there are many abbreviations that are not mentioned in their full name.

-The author also should mention the cytotoxic effects of those nanoparticles towards normal not cancer cells only (if present) or safety towards normal cells, because those NPs would reach both normal and cancer cells in vivo and might cause toxicity to their DNA.

- In the current state, there are more typographical errors. Therefore, the authors are advised to recheck the whole manuscript for improving the language and structure carefully.

Author Response

Author reply to the reviewer comment.

Comments and Suggestions for Authors

Some specific comments are as follows:

Page 2, Line 60…..…. small cell lung cancer (SCLC) and non-small cell lung cancer (NSCLC)……… add the full name for the first time before their abbreviations.

Reply: I have done a correction.

Page 2, Line 63…..…. according to the WHO data for 2020……..Where is its’ reference?

Reply: Dear reviewer, I have included a reference.

Page 3, Line 88…..…. the transporters……The (begin with Capital letter).

Reply: I have done a correction.

Page 3, Lines 88-90……need ref., is it ref [10]?

Reply: I have included a reference.

Page 3, Line 93…..…. References number [9] and [10] have no relation to their cited paragraph about cisplatin, paclitaxel, and doxorubicin and their toxicity….Please check and add the right refs.

Reply: I have corrected a reference.

[9] Huang, Y.; Hu, X.; Zhao, H.; He, D.; Li, Y.; Yang, M.; Yu, Z.; Li, K.; Zhang, J. Composite alkali polysaccharide supramolecular nanovesicles improve biocharacteristics and anti-lung cancer activity of natural phenolic drugs via oral administration. Int J Pharm 2020, 573, 118864, doi:10.1016/j.ijpharm.2019.118864.

[10] Jeannot, V.; Mazzaferro, S.; Lavaud, J.; Vanwonterghem, L.; Henry, M.; Arboléas, M.; Vollaire, J.; Josserand, V.; Coll, J.L.; Lecommandoux, S.; et al. Targeting CD44 receptor-positive lung tumors using polysaccharide-based nanocarriers: Influence of nanoparticle size and administration route. Nanomedicine 2016, 12, 921-932, 761 doi:10.1016/j.nano.2015.11.018.

I thought there is something wrong in citation! or there is a mistake in references numbering? Please check the reference list with their citation within the text.

Reply: I have done a correction.

Page 3, Lines 109-113…..…. References number [13] and [14] have no relation to their cited paragraph ….Please check and add the right refs.

Reply: Dear reviewer, I have corrected a references.

Page 3, Lines 116-120…..…. References number [16] and [17] have no relation to their cited paragraph ….Please check and add the right refs.

Reply: Dear reviewer, I have corrected a references.

Page 3, Lines 121,122……need reference

Reply: I have included a reference.

 Page 3, Line 122…..…. COPD ……… add the full name for the first time before its abbreviation.

Reply: I have done a correction.

Page 4, Lines 124-127…..…. Reference number [20] have no relation to its cited paragraph ….Please check and add the right refs.

Reply: I have done a correction in reference.

Page 6…..Table 1, where are the references of the advantages of those referred Polysaccharides?

Reply: Dear reviewer, as per your suggestion, I have included a references in table 1.

Page 7, Line 160…..…. TPP……… add the full name for the first time before its abbreviation.

Reply: I have done a correction.

Page 7, Line 167…..…. they are…….Capital (They)

Reply: I have done a correction.

Page 7, Line 171…..…. EGFR……… add the full name for the first time before its abbreviation.

Reply: I have done a correction.

Page 7, Line 172……………Matte Kasi Viswanath et al.? This reference was not found in the reference partition. Is it Viswanadh [32]?

Reply: It was a typo error, it is a Viswanadh. I have done a correction in  revised manuscript.

Page 7, Line 172……………QbD method……… add the full name for the first time before its abbreviation.

Reply: I have done a correction.

Page 7, Line 183…………… DevarajBharathi et al………………… Bharathi et al. [33]

Reply: Dear reviewer, due to mistake in sentence, I have deleted a sentence.

Page 8, Line 196…………… AryaN et al…………. Arya et al. [34]

Reply: I have done a correction.

Page 8, Line 199…………… Tx and TPT……… add the full name for the first time before its abbreviation.

Reply: I have done a correction.

Page 8, Line 211…………… Giuseppe Cirillo and colleagues…………. Giuseppe Cirillo and colleagues

[35]

Reply: I have done a correction.

Page 9, Line 232………… Yun-qiu Miao et al……….Reference number[]

Reply: I have included a reference.

Is there any explained mechanism for Chitosan based nanomedicines toxicity toward cancer cells?

Reply: Dear reviewer, I have included a study in revised manuscript.

Figure 2: in page 10……Those figures are from article [33], Did the author obtain permission from the copyright holder to publish under the CC-BY license? According to Publication Ethics Statement of MDPI

https://www.mdpi.com/journal/information/instructions

Reply: Dear reviewer, yes I have taken a permission from a journal. I have attached all permission certificate with this reply.

Page 10, Line 281……………………. HA……. add the full name for the first time before its abbreviation.

Reply: I have done a correction.

Page 11, Line 301………… VictorJeannot et al…………….. Jeannot et al [Reference number]

Reply: I have done a correction.

Page 12, Line 338………… PoonamParashar et al…………… Parashar et al. [Reference number], Lines 338-339 should be in Lowercase sentence.

Reply: I have done a correction.

Figure 3: in page 10……Those figures are from article [43], Did the author obtain permission from the copyright holder to publish under the CC-BY license? According to Publication Ethics Statement of MDPI

https://www.mdpi.com/journal/information/instructions

Reply: Dear reviewer, yes I have taken a permission from a journal. I have attached all permission certificate with this reply.

Page 13, Lines 383-385……….. Laminaria japonica, Macrocystis pyrifera, Ascophyllum nodosum, Laminaria digitata, and Laminaria Hyperborea………………Should be Italics

Reply: I have done a correction.

Page 16, Lines 459-460 ……. RhamnogalacturonaI About 20–35% of the total pectin mass is represented by one component of a pectin…………………………. RhamnogalacturonaI (about 20–35% of the total pectin mass) is represented by one component of a pectin

Reply: I have done a correction

Page 16, Line 467…………… YunLi et al………….. Li et al. [Reference number]

Reply: I have done a correction

Page 16, Line 474…………… (%).In every case,…………………….need space

Reply: I have done a correction

Page 16, Lines 487-488………………..Pectin, Guar Gum, and Zinc Oxide Nanocomposite for Apoptotic Cell Death Induction 487 Lung Adenocarcinomas was created by Indu Hira et al……..This reference is not found in the reference part

Reply: Dear reviewer, I have corrected a reference.

Page 18, Lines 517……… Bacteroidesthetaiotaomicron…………correct Bacteroides thetaiotaomicron

Reply: I have done a correction

Page 18, Lines 519-521…………………….. Recently, we produced chondroitin sulfate-based nano………..We??? I didn’t find reference for author’s work… please clarify or correct.

Reply: Dear reviewer, due to mistake in sentence, I have deleted this sentence from revised manuscript.

Page 18, Line 526……………… Adr,…………………….. add the full name for the first time before its abbreviation.

Reply: I have done a correction

-Please check references well inside the text and there are many abbreviations that are not mentioned in their full name.

Reply: Dear reviewer, I have corrected a reference and abbreviations.

-The author also should mention the cytotoxic effects of those nanoparticles towards normal not cancer cells only (if present) or safety towards normal cells, because those NPs would reach both normal and cancer cells in vivo and might cause toxicity to their DNA.

Reply: I have done a correction in revised manuscript.

- In the current state, there are more typographical errors. Therefore, the authors are advised to recheck the whole manuscript for improving the language and structure carefully.

Reply: Dear reviewer, I have revised a manuscript and corrected a typo error and grammatical mistakes.

Round 2

Reviewer 1 Report

The author responded and included the majority of my suggestions into the new version.

The following specific comments still need to be corrected:

Line 60: You didn't understand my previous comments when I said that cardiovascular disease is the leading cause of death in the world. I didn't mean for you to write it because it's irrelevant to your manuscript but correct it by saying that cancer is one of the leading causes of death in the world, not the leading one.

Line 330: Use only one of the 2 terms uptake or internalization, not both.

In addition, you did not comment and updated the conclusion section in response to my proposal to improve it. I would have appreciated it if you enhanced this section by including more questions and discussions regarding the current status of research in this field, or what is missing in the literature, I suggested some specific topics that you might have covered:

·         The general understanding of how polysaccharide-based nanoparticles are internalized by cells.

·         The availability of biodistribution and clearance studies (studies in vivo animals and humans).

·         The stability of polysaccharide-based nanoparticles in vivo.

·         Discuss the polysaccharide-based NPs synthesis process (difficulties, cost) in comparison to other nanomaterials.

·         The possibility of functionalization of such nanoparticles using other specific ligands (peptides, proteins, antibodies, etc.) to target and treat lung cancers.

Author Response

Author reply to the reviewer comments

Comments and Suggestions for Authors

The author responded and included the majority of my suggestions into the new version.

The following specific comments still need to be corrected:

Line 60: You didn't understand my previous comments when I said that cardiovascular disease is the leading cause of death in the world. I didn't mean for you to write it because it's irrelevant to your manuscript but correct it by saying that cancer is one of the leading causes of death in the world, not the leading one.

Reply: Dear reviewer, many thanks for your comments. I am sorry for not understanding your comment. Now I have corrected a sentence.

Line 330: Use only one of the 2 terms uptake or internalization, not both.

Reply: Dear reviewer, I have corrected a sentence.

In addition, you did not comment and updated the conclusion section in response to my proposal to improve it. I would have appreciated it if you enhanced this section by including more questions and discussions regarding the current status of research in this field, or what is missing in the literature, I suggested some specific topics that you might have covered:

  • The general understanding of how polysaccharide-based nanoparticles are internalized by cells.
  • The availability of biodistribution and clearance studies (studies in vivo animals and humans).
  • The stability of polysaccharide-based nanoparticles in vivo.
  • Discuss the polysaccharide-based NPs synthesis process (difficulties, cost) in comparison to other nanomaterials.
  • The possibility of functionalization of such nanoparticles using other specific ligands (peptides, proteins, antibodies, etc.) to target and treat lung cancers.

Reply: Dear reviewer, many thanks for your suggestion. As per your suggestion I have modified a conclusion section.